# An Exploratory Study in Non-Professional Football on the Perception of Stakeholders about the New Working Professional Profile of Sports Kinesiologist

**DOI:** 10.3390/ijerph192315839

**Published:** 2022-11-28

**Authors:** Gaetano Raiola, Tiziana D’Isanto, Francesca D’Elia, Gaetano Altavilla

**Affiliations:** 1Department of Political and Social Studies, University of Salerno, 84084 Fisciano, Italy; 2Department of Human, Philosophical and Education Sciences, University of Salerno, 84084 Fisciano, Italy

**Keywords:** sports science and exercise, trainers, sports law, professional sport profile

## Abstract

In Italy, recent amendments to Legislative Decree n. 36 of 28 February 2021, on sports work, may have made the application of the reform by stakeholders unclear, with the risk of generating further confusion among them. One of the most critical points concerns the possible equivalencies to the professional profile of the kinesiologist, which would be illegitimately recognized even for a different level of education, contrary to the requirements of the European qualification framework. The aim of the study was to understand the perceptions of stakeholders in the world of non-professional football regarding recent legislative provisions. A survey, divided into two sections, was administered to 112 presidents and 112 trainers of non-professional football associations of the province of Salerno. The first section presents five items common for both presidents and trainers, which seek to probe stakeholders’ perceptions of the enjoyment, appropriateness, usefulness, and scientificity of kinesiologists. The second section presents five differentiated items. Validity and reliability were calculated. A chi-square analysis (χ^2^) was performed to test the independence within and between-subjects (trainers and presidents) on their perceptions about the new working professional profile of sports kinesiologist. From the results, it was possible to appreciate a discordance of opinion among stakeholders. Although the majority of presidents and trainers are in favour of introducing such a professional profile (*p* < 0.05), contradictions emerge concerning the contribution the new professional profile can make in practice (*p* > 0.05). The perceptual contradictions found among stakeholders’ responses demonstrate how the complexity of recent regulatory provisions regarding possible equivalencies to the title of kinesiologist have inevitably generated further confusion among stakeholders.

## 1. Introduction

The day of 28 February 2021, marked a turning point for exercise and sport sciences in Italy since the reform of sports workers was enacted through Legislative Decree n. 36 on reorganizing and reforming provisions on professional and amateur sports bodies and sports labour, implementing article 5 of Law n. 86 of 8 August 2019 [1]. This reform contains significant innovations, including the recognition of the working professional profiles of the basic kinesiologist, the preventive and adapted physical activities kinesiologist, the sports kinesiologist, and the sports manager in Italy as also in a large part of the world [2,3,4,5]. The reasons such new working professionals’ profiles are legally recognized are to pursue the proper conduct of physical and sports activities, well-being protection, and the promotion of healthy lifestyles.

Of particular interest is the sports kinesiologist as defined by article 41, comma 4 of the decree, whose exercise of activity has to do with: “the planning, coordination and technical direction of athletic preparation activities in the competitive sphere up to the highest levels of competition for sports associations and clubs, sports promotion bodies, institutions and specialized centres, personalized physical and technical preparation aimed at individual and team competitiveness”. Those with a master’s degree in sports science and techniques (LM68) will be eligible for a profession whose practice is exclusive [6]. All this will bring contractual benefits for this professional profile since labour contracts will have to be regulated with the regulations inherent in the national collective agreements for individual categories of workers. In addition, comma 6 of article 41 also deals with possible equivalencies to the title of kinesiologist, where the provision states that “criteria for the recognition of equivalent qualifications for professional practice should be established”. Another particularly new element of the decree is described in article 42, comma 1, which specifies that: “Courses and sports activities offered within gyms, centres and sports facilities of any kind, against payment of fees in any capacity, including in the form of membership fees, must be conducted under the coordination of a kinesiologist or trainer of the specific discipline”. Article 42, comma 4 also specifies that: “National Sports Federations, Associated Sports Disciplines or Sports Promotion Bodies recognized by the National Olympic Committee, are exempted from the obligation to have the kinesiologist as coordinator of the sports activities they regulate within the sports centres”.

Following further very recent amendments to the recognition of sports qualifications of National Sports Federations (NSF), it has become more difficult to understand the legislative innovation because it equates the master’s degree in exercise and sport sciences with the NSF trainer as a new professional profile [7]. This condition violates the existing legislative provision of the European qualification framework (EQF) for the free movement of degree holders in the European Union (EU) in the field of exercise and sport sciences according to the guidelines on physical activity and sedentary lifestyle of the World Health Organization [8].

This problem, for which it is necessary to understand stakeholders’ perceptions, is a complex issue determined by such legislative provisions. Notably, some of the corrections made to the decree may not clarify the reform application to stakeholders, with the risk of increasing further confusion. One of the most critical points concerns the possible equivalencies to the professional profile of the kinesiologist, which, according to article 42, paragraph 1, would be illegitimately recognized even for a different level of training, contrary to what the EQF requires. Under this EU legislative provision, any equivalence between degrees of different weight and value, such as a master’s degree and certificates of higher technical specialization courses issued by the NSF, would be difficult to sustain. Another critical element is expressed in article 42, paragraph 4(a) regarding the exemption of competitive sports activities from the obligation of the coordinator’s presence. This could generate discrimination between sports clubs that do not engage in competitive activities and are not affiliated with an NSF and sports clubs that, although they do not engage in competitive activities, like the former, are instead affiliated with an NSF.

Given these problematic issues, it would be useful to make original use of survey methods, such as those on perception and awareness [9], to methodologically finalize cognitive inquiry with similar tools. The aim of the study was to understand and compare the perceptions of stakeholders, particularly presidents and trainers, in the world of non-professional football regarding recent legislative provisions and corrective provisions reorganizing and reforming sports labour provisions. The intention was to check whether presidents and trainers were aware that the inclusion of a kinesiologist profile in their team could significantly improve the quality of the sports club.

## 2. Materials and Methods

### 2.1. Design and Participants

The investigation was conducted through an online survey targeting a sample characterized by 112 presidents (mean ± standard deviation = 45 ± 2.99 years old) and 112 trainers (33 ± 3.52 years old) in non-professional football associations located in the province of Salerno (Italy). They carry out competitive activities and are affiliated with the Italian Football Federation. The numerosity and significant representativeness of the sample at the territorial level make it suitable for professional and scientific discussion.

### 2.2. Data Collection

After choosing the target population, data collection was carried out taking into account both methodological implications and available economic and human resources. On this basis, it was decided to administer a survey, the writing of which was based on the conceptual dimensions and related indicators identified during the definition of the research objective. Some studies in the literature have demonstrated the validity of the survey in detecting the perceptions of stakeholders [10,11,12].

### 2.3. Validity Procedure

Validity refers to the fact that an instrument measures exactly what it purports to measure [13]. In this study, face and content validity were assessed. Face validity refers to the conciseness of the items of the instrument concerning clarity, brevity, and completeness [14]. Content validity refers to the degree to which items in an instrument reflect all aspects of the construct [15] and is based on the judgement of a group of experts in a specific area of interest. To guarantee these two types of validity, the surveys were revised by two study groups. The first group were survey construction experts who adapted the structure of the surveys as they saw fit. The second group were sports management experts who assessed whether the questions correctly captured the topic. Items with content validity index (CVI) greater than 0.78 were included in the final instrument. The final version of both surveys consisted of two thematic sections. The first section presented five items common for both presidents and trainers, which sought to probe stakeholders’ perceptions of the enjoyment, appropriateness, usefulness, and scientificity of kinesiologists. The second section presented five differentiated items. Presidents were asked to make an overall assessment of the work and knowledge of the trainers registered with their football associations. Trainers, on the other hand, were asked to self-assess themselves, judging their technical, methodological, and scientific knowledge. The survey administered to the presidents is shown in Table 1. 

The survey administered to the trainers is shown in Table 2. 

### 2.4. Reliability Procedure

Reliability is the ability to reproduce a result consistently across time and space or from different observers [13]. The reliability criteria calculated in this study were stability, which measures how similar the results measured at two different points in time are through the test–retest with a sample of at least 50 subjects and the calculation of the intraclass correlation coefficient (ICC) [16], and internal consistency, which shows whether all items of an instrument measure the same characteristic through Cronbach’s α coefficient [17].

### 2.5. Statistical Analysis

To validate the surveys, we first assessed its internal consistency through Cronbach’s α and associated 95% confidence intervals (CI). A Cronbach’s α of 1 indicated perfect reliability, with a cut-off of 0.70 indicating an acceptable internal consistency [18].

Then, we assessed the test–retest reliability by administering the surveys after 1 month to a sub-sample of 50 presidents and 50 trainers who agreed to be contacted again about the study [19]. The ICC was interpreted using the criteria suggested by Portney and Watkins [20] as poor reliability (ICC ≤ 0.50), moderate reliability (ICC 0.50–0.75), good reliability (ICC 0.75–0.90), and excellent reliability (ICC ≥ 0.90).

A chi-square analysis (χ^2^) was performed to test the independence within and between subjects (trainers and presidents) on their perceptions about the new working professional profile of sports kinesiologist. Significance was set at *p* < 0.05. Data analyses were performed using the Statistical Package for Social Science software (IBM SPSS Statistics for Windows, version 25.0, IBM, SPSS Inc., Armonk, NY, USA).

## 3. Results

### 3.1. Data Quality Check

The internal consistency of the survey for presidents was excellent (Cronbach’s α coefficient [95% CI] 0.92 [0.89–0.94]; *p* < 0.000). In addition, the survey for trainers had a good internal consistency (Cronbach’s α coefficient [95% CI] 0.85 [0.81–0.89]; *p* < 0.000). The test–retest reliability of the survey for presidents ranged from moderate to excellent, while that for trainers was from good to excellent. A detailed description is shown in Table 3.

### 3.2. Chi Square Associations

Chi-square showed two significant associations between presidents’ and trainers’ perceptions, specifically regarding their knowledge of the professional profile of kinesiologist (χ^2^ = 364; *p* = 0.05) and the most suitable figure among the three kinesiologist profiles to fill the technician role (χ^2^ = 18.2; *p* = 0.00). A detailed description is shown in Table 4.

Chi-square showed two significant associations among presidents’ perceptions, specifically regarding being completely in favour of the introduction of the kinesiologist profile into their technical staff because they intuit the kinesiologist role (χ^2^ = 7.76; *p* = 0.00) and also because they expect that the kinesiologist will greatly improve the training quality (χ^2^ = 8.93; *p* = 0.00). A detailed description is shown in Table 5.

Chi-square showed two significant associations among trainers’ perceptions. The first is about being completely in favour of the introduction of the kinesiologist profile into the technical staff because they concretely intuit the kinesiologist role (χ^2^ = 7.84; *p* = 0.00). The second is about the trainers’ perception that the kinesiologist will greatly improve the training quality and their awareness of the importance of adapting training according to the individual development stages across age groups (χ^2^ = 7.84; *p* = 0.05). A detailed description is shown in Table 6.

## 4. Discussion

The results made it possible to appreciate some discordance of opinion among stakeholders concerning the current legislative provisions. Although the majority of presidents and trainers were in favour of introducing such a professional profile, some contradictions emerged concerning the contribution this profile could make in practice. An initial difference in perception between presidents and trainers, as shown in Table 3, was found in the first question, which asked stakeholders whether they were familiar with the new professional profile introduced by the Sports Reform Act of 2021 (*p* = 0.05). The second difference in perception was found in the third question asking stakeholders to identify the most suitable professional profile to fill the role of technician (*p* = 0.00). No further differences in perception were found for the remaining common items. Table 4 shows the differences in perceptions found among the presidents. In this case, two significant differences in perceptions were found. Presidents unaware of what the kinesiologist might be involved in were not entirely sure about introducing such a profile into their football associations. In addition, while presidents who were not wholly in favour of introducing kinesiologists to their football associations believed that this profile would not be able to improve further the quality of football training offered. In contrast, all those presidents who were entirely in favour of introducing kinesiologists to their football associations believed that the quality of football training could certainly be significantly increased. For the remaining items, no differences in perceptions were identified. Finally, Table 5 shows the two differences in perceptions found among trainers. In this case, trainers who said they knew concretely what the kinesiologist did were in favour of introducing him as a staff member compared with those who did not know what he did concretely (*p* = 0.05). In addition, those trainers who said they had good knowledge of the developmental stages of the individual in the various age groups agreed that the kinesiologist could significantly improve the quality of football training (*p* = 0.00). Similarly, those who stated that they had fair/sufficient knowledge regarding this topic were fairly/poorly convinced that the kinesiologist could improve the quality of football training.

It is clear from the responses that several non-professional football associations currently have more former athletes or federally licensed trainers in their technical staff than kinesiologists. In this sense, article 27 of the recent legislative decree, intervening on article 41 of Legislative Decree n. 36 of 2021, clarifies the respective professional competencies: kinesiologists must deal with the movement of the body of those who perform motor activities; trainers of specific sports disciplines must deal with the performance of competitive sports activities [7]. With Official Statement n. 1 of 1 July 2020, the FIGC entrusted the technical conduct of youth teams exclusively to trainers qualified by its education system, thus excluding master’s graduates in exercise and sport sciences [21]. In application of the amendments made in 2018 to Part II of the Technical Sector Regulations, the FIGC has established that sports clubs that carry out youth and school sector activities must use at least one trainer with UEFA federal qualification issued by the Technical Sector for each age category of players [22]. Thus emerges the legislature’s desire to recognize equal legal weight between trainers of sports federations and master’s graduates in exercise and sport sciences. In such a complex scenario, there are critical issues in the contents of the decree that do not allow for a correct application of the rules, especially regarding the possible equivalence between the title of kinesiologist and the specific professional qualification.

These critical issues have also been highlighted by the Conference of Autonomous Regions and Provinces (CARP), which recently called for a thorough evaluation of the implementation of the new regulations [23]. As can be appreciated from the results of this study, at the moment, it is still unclear what concrete effects this reform will have on the professional future of kinesiologists but more importantly on the effective health protection of citizens who participate in sports. Encouraging proper physical and sports practice in citizens of all ages is important given its benefits, especially in the wake of the COVID-19 pandemic [24,25,26]. In order to achieve maximum benefits, the technical-practical skills and expertise of the kinesiologist are crucial. Greater clarity is required from the institutions concerning the profile of the sports practitioner in the hope that proper attention will be paid to the opinion expressed by the territory (CARP) and that the suggested changes will help to give due value to the more than 100,000 sports science graduates who have been trained in these 20 years.

## 5. Conclusions

This study showed how the complexity and contradictory nature of recent regulatory provisions regarding possible equivalencies to the title of kinesiologist inevitably increased further confusion among stakeholders. There is a need for implementing provisions declining applications for different specific cases. Regarding the method of the study and considering the sampling limitations of the study, the lack of demographic data, and the primitive wording of the surveys, it is necessary to replicate it in order to provide useful elements to the legislator.

## Figures and Tables

**Table 1 ijerph-19-15839-t001:** Survey administered to the presidents.

Q1	Do you know the professional profile of the kinesiologist defined in the Sports Reform Act of 2021? (a) Yes (b) No (c) Fairly
Q2	Do you concretely intuit what the kinesiologist may be involved in? (a) Yes (b) No (c) Fairly
Q3	Which of the following professional profiles defined by the Sports Reform Act of 2021 is best suited to fill the technician role? (a) Basic kinesiologist (b) Sports kinesiologist (c) Preventive and adapted physical activities kinesiologist
Q4	How much do you expect the kinesiologist will improve the quality of training in your club? (a) A lot (b) Fairly (c) Not a lot
Q5	Are you in favour of introducing the profile of the kinesiologist into the club’s technical staff alongside the current coach? (a) Strongly agree (b) Somewhat agree (c) Somewhat disagree
Q6	What cultural and technical qualifications should have a football trainer? (a) Italian National Olympic Committee (CONI)–Italian Football Federation (FIGC) qualification (b) Union of European Football Associations (UEFA) license (c) Master’s degree in exercise and sport sciences (d) Apprenticed trainee (e) Former athlete
Q7	How do you think your trainers carry out their profession? (a) Passionately (b) Competently (c) By habit (d) For fun
Q8	What kind of technical preparation are your trainers able to give their teams? (a) Good (b) Discreet (c) Sufficient (d) Insufficient
Q9	What kind of behaviour do your trainers engage in during training? (a) Authoritarian (b) Impulsive (c) Permissive (d) Positive
Q10	How do you rate your trainers’ knowledge of the developmental stages of growth, motor development, and sensitive periods of development? (a) Good (b) Discreet (c) Sufficient (d) Insufficient

**Table 2 ijerph-19-15839-t002:** Survey administered to the trainers.

Q1	Do you know the professional profile of the kinesiologist defined in the Sports Reform Act of 2021? (a) Yes (b) No (c) Fairly
Q2	Do you concretely intuit what the kinesiologist may be involved in? (a) Yes (b) No (c) Fairly
Q3	Which of the following professional profiles defined by the Sports Reform Act of 2021 is best suited to fill the technician role? (a) Basic kinesiologist (b) Sports kinesiologist (c) Preventive and adapted physical activities kinesiologist
Q4	How much do you expect the kinesiologist will improve the quality of training in your club? (a) A lot (b) Fairly (c) Not a lot
Q5	Are you in favour of introducing the profile of the kinesiologist into the club’s technical staff alongside the current trainer? (a) Strongly agree (b) Somewhat agree (c) Somewhat disagree
Q6	What cultural and technical qualifications do you possess? (a) CONI-FIGC qualification (b) UEFA license (c) Master’s degree in exercise and sport sciences (d) Apprenticed trainee (e) Former athlete
Q7	Are you able to correctly evaluate and select the different means and methods of training? (a) Yes (b) No (c) Enough (d) A little
Q8	How do you rate your knowledge concerning individual development stages across age groups? (a) Good (b) Discreet (c) Sufficient (d) Insufficient
Q9	How do you rate your interpersonal, organizational/management, and programming soft skills? (a) Good (b) Discreet (c) Sufficient (d) Insufficient
Q10	How do you rate your skills in being able to plan, coordinate, and direct physical and athletic training activities? (a) Good (b) Discreet (c) Sufficient (d) Insufficient

**Table 3 ijerph-19-15839-t003:** Test re-test reliability.

Survey for Presidents	Test–Retest Reliability		Survey for Trainers	Test–Retest Reliability	
Variable	ICC (95%CI)	*p*	Variable	ICC (95%CI)	*p*
Q1	0.95 (0.92–0.97)	<0.000	Q1	0.91 (0.84–0.94)	<0.000
Q2	0.96 (0.93–0.97)	<0.000	Q2	0.93 (0.88–0.96)	<0.000
Q3	0.66 (0.41–0.81)	<0.000	Q3	0.97 (0.95–0.98)	<0.000
Q4	0.97 (0.96–0.98)	<0.000	Q4	0.94 (0.90–0.96)	<0.000
Q5	0.91 (0.85–0.95)	<0.000	Q5	0.94 (0.91–0.97)	<0.000
Q6	0.93 (0.88–0.96)	<0.000	Q6	0.81 (0.67–0.89)	<0.000
Q7	0.91 (0.84–0.95)	<0.000	Q7	0.97 (0.95–0.98)	<0.000
Q8	0.86 (0.74–0.92)	<0.000	Q8	0.93 (0.88–0.96)	<0.000
Q9	0.97 (0.95–0.98)	<0.000	Q9	0.91 (0.84–0.95)	<0.000
Q10	0.87 (0.78–0.92)	<0.000	Q10	0.86 (0.75–0.92)	<0.000

**Table 4 ijerph-19-15839-t004:** Differences in perception between presidents and trainers.

		Presidents	Trainers	χ^2^	*p*
Q1. Do you know the professional profile of the kinesiologist defined in the Sports Reform Act of 2021?	No	52	38	3.64	0.05
	Yes	60	74		
Q2. Do you concretely intuit what the kinesiologist may be involved in?	Fairly	36	32	1.05	0.59
	No	20	26		
	Yes	56	54		
Q3. Which of the following professional profiles defined by the Sports Reform Act of 2021 is best suited to fill the technician role?	Preventive and adapted physical activities kinesiologist	22	24	18.2	0.00
	Basic kinesiologist	10	34		
	Sports kinesiologist	80	54		
Q4. How much do you expect the kinesiologist will improve the quality of training in your club?	Fairly	38	38	1.37	0.50
	A lot	60	54		
	Not a lot	14	20		
Q5. Are you in favour of introducing the profile of the kinesiologist into the club’s technical staff alongside the current trainer?	Somewhat agree	38	44	0.92	0.63
	Strongly agree	54	52		
	Somewhat disagree	20	16		

**Table 5 ijerph-19-15839-t005:** Differences in perception found among presidents.

			Q2. Do you concretely intuit what the kinesiologist may be involved in?				
			Fairly	No	Yes	χ^2^	*p*
Q5. Are you in favour of introducing the profile of the kinesiologist into the club’s technical staff alongside the current trainer?	Somewhat agree		36	2	0	7.76	0.00
	Strongly agree		0	0	54		
	Somewhat disagree		0	18	2		
			Q4. How much do you expect the kinesiologist will improve the quality of training in your club?				
			Fairly	A lot	Not a lot	χ^2^	*p*
Q5. Are you in favour of introducing the profile of the kinesiologist into the club’s technical staff alongside the current trainer?		Somewhat agree	38	0	0	8.93	0.00
		Strongly agree	0	54	0		
		Somewhat disagree	0	6	14		

**Table 6 ijerph-19-15839-t006:** Differences in perception found among trainers.

		Q2. Do you concretely intuit what the kinesiologist may be involved in?				
		Fairly	No	Yes	χ^2^	*p*
Q5. Are you in favour of introducing the profile of the kinesiologist into the club’s technical staff alongside the current trainer?	Somewhat agree	32	12	0	7.84	0.00
	Strongly agree	0	0	52		
	Somewhat disagree	0	14	2		
		Q4. How much do you expect the kinesiologist will improve the quality of training in your club?				
		Fairly	A lot	Not a lot	χ^2^	*p*
Q8. How do you rate your knowledge concerning individual development stages across age groups?	Good	0	30	0	7.84	0.05
	Discreet	0	24	20		
	Sufficient	36	0	0		
	Insufficient	2	0	0		

## Data Availability

Not applicable.

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
