# Peer review of "An Exploratory Study in Non-Professional Football on the Perception of Stakeholders about the New Working Professional Profile of Sports Kinesiologist"

_ijerph, 2022, doi:10.3390/ijerph192315839_

Round 1
Reviewer 1 Report
Overall, the paper was presented well. the writing is good. the structure is clear. however, improvements are needed:
1. title is too long, please revise the title of this paper
2. research question is unclear to readers, the authors need to clearly state what are their research questions, and how the research questions are answered.
3. The survey questions and formats need to be carefully thinking, it has different scales, and there is no room for the participants to express their perceptions/ideas further, open-ended questions are also needed.
Author Response
LETTER TO REVIEWERS
Dear Reviewers,
thank you for reviewing our manuscript. We appreciate it. We have followed your suggestions point by point to improve the manuscript quality, according to our possibilities. Changes have been made to the full text using word tracking to detect changes immediately.
AUTHORS (A)
REVIEWER 1 (R1)
R1: Overall, the paper was presented well. the writing is good. the structure is clear. however, improvements are needed: title is too long, please revise the title of this paper
A: Thanks for your advice. We proceeded with the reduction of the title.
R1: research question is unclear to readers, the authors need to clearly state what are their research questions, and how the research questions are answered.
A: Thanks for your advice. We further specified the aim of the study:
- to investigate the perceptions of two groups of stakeholders on the new sports reform.
- to test both intra- and inter-group independence regarding the perceptions on the new sports reform (H0 = independence, H1 = dependence)
R1: The survey questions and formats need to be carefully thinking, it has different scales, and there is no room for the participants to express their perceptions/ideas further, open-ended questions are also needed.
A: Thanks for your advice. The format was studied by a group of survey experts, while open questions were not included because they do not allow the comparison of data through statistics. Validity and reliability give us positive results. Overall, we tried to improve tha manuscript quality by including additional and specific information about the surveys and rewriting sentences more clearly.
Thanks for your time.
Reviewer 2 Report
The study lacks more robustness. For example, consideration should be given to the age, academic qualifications and length of service of Presidents and Coaches, as well as the coach's degree/level.
The statistical treatment should be revised according to the specification of the variables mentioned above.
Explain better how the survey instrument was created and how it was validated.
The conclusions have a part written in Italian.
Author Response
LETTER TO REVIEWERS
Dear Reviewers,
thank you for reviewing our manuscript. We appreciate it. We have followed your suggestions point by point to improve the manuscript quality, according to our possibilities. Changes have been made to the full text using word tracking to detect changes immediately.
AUTHORS (A)
REVIEWER 2 (R2)
R2: The study lacks more robustness. For example, consideration should be given to the age, academic qualifications and length of service of Presidents and Coaches, as well as the coach's degree/level.The statistical treatment should be revised according to the specification of the variables mentioned above.
A: Thanks for your advice. Unfortunately, the absence of demographic variables is a limitation of our study (We proceeded to write this down in the text) and could be considered for future studies. Only age and location were information available to us. However, our aim was simply to investigate perceptions regarding the new sports reform and to highlight differences in perceptions between two types of stakeholders working together (presidents and coaches). For future studies, we could investigate the influence of demographic variables on perceptions of whether or not to introduce a kinesiologist into one's team.
R2: Explain better how the survey instrument was created and how it was validated.
A: Thanks for your advice. We have specified well the validation and reliability procedure used.
R2: The conclusions have a part written in Italian.
A: We apologise. We have translated it.
Thanks for your time.
Round 2
Reviewer 2 Report
We believe that the improvements introduced are sufficient for the article to be accepted, however we would like to recommend that in future studies on this topic, they should take into account a greater specification of the variables related to the sample so that it is possible to better identify and understand the opinions of presidents and trainers.
Author Response
Dear Reviewer,
thank you for your suggestions. We are in complete agreement with you. In future studies, we will specify the demographic characteristics of the sample surveyed.